# Characterizing the services provided by family physicians in Ontario, Canada: A retrospective study using administrative billing data

David W. Savage[1], Arunim Garg[2], Salimur Choudhury[3], Roger Strasser[1], Robert Ohle[3], Vijay Mago [4] *

1 Clinical Sciences Division, NOSM University, Thunder Bay, Ontario, Canada, 2 Department of Computer Science, Lakehead University, Thunder Bay, Ontario, Canada, 3 School of Computing, Queen's University, Kingston, Ontario, Canada, 4 School of Health Policy & Management, York University, Toronto, Ontario, Canada

* vmago@yorku.ca

## Abstract

Family physicians in Ontario provide most of the primary care to the healthcare system. However, given their broad scope of practice, they often provide additional services including emergency medicine, hospital medicine, and palliative care. Understanding the spectrum of services provided by family physicians across different regions is important for health human resource planning (HHRP). We investigated the services provided by family physicians in Ontario, Canada using a provincial physician database and administrative physician billing data from 2017. Billing codes were used to define 18 general services that family physicians may provide. We then evaluated variation in the services provided by different physicians based on the physicians' geographic location (north-urban, north-rural, south-urban, and south-rural) and career stage (i.e., years in practice). Ontario had 14,443 family physicians in 2017, with most practicing in urban communities in southern Ontario and only 6.5% practicing in any setting in northern Ontario. In general, rural physicians provided a greater range of services than their urban colleagues. Their practices most often included clinic medicine, mental health services, emergency medicine, palliative care, and hospital medicine. Physicians in urban southern Ontario and those at a more advanced career stage were more likely to provide a narrower range of services. Overall, our findings have the potential to shape HHRP, medical education curriculum development, and clinical services planning in Ontario and elsewhere. Moreover, our results provide policy- and decision-makers with a basis for integrating knowledge of the specific clinical services delivered by family physicians into their future planning, with the goal of ensuring a fit-for-purpose workforce able to meet community healthcare needs.

## Introduction

Health human resource planning (HHRP) is the process of ensuring that the right number of physicians, with the right skills, are available to a population in the right place and at the right

---

---

**Data Availability Statement:** The dataset from this study is held securely in coded form at ICES. While

data sharing agreements prohibit ICES from making the dataset publicly available, access may be granted to those who meet pre-specified criteria for confidential access, available at www.ices.on.ca/DAS. The full dataset creation plan and underlying analytic code are available from the authors upon request, understanding that the computer programs may rely upon coding templates or macros that are unique to ICES and are therefore either inaccessible or may require modification. Also, we would like to mention that other researchers would be able to access these data in the same manner as the authors and that the authors did not have any special access privileges that others would not have.

**Funding:** RS is a co-investigator with DS as the principal investigator on a grant from the Northern Ontario Academic Medical Association (NOAMA) Clinical Innovation Opportunities Fund (C-18-19). Additional funding to SC from the Natural Sciences and Engineering Research Council (NSERC) of Canada supported this work.

**Competing interests:** None

time [1]. HHRP is essential for determining and allocating the resources needed to meet each community's healthcare needs. Moreover, strong primary care systems have been shown to improve overall population health through health prevention, management of chronic disease, improved patient outcomes and a reduction in health system costs [2]. Primary care is essential to public health since it addresses health inequities faced by underserved populations and by coordinating care across different providers [3]. The benefits of primary care can be attributed to several factors, but one of the strongest determinants of patient outcomes is the accessibility of required services [2].

In Canada, family physicians provide much of the primary care and a spectrum of services to their patients [4, 5]. Family physicians working in rural settings typically engage in an even broader spectrum of professional practice than their counterparts in urban environments [6]; largely because rural areas have fewer physicians per capita but service a similar range of health care needs [4, 7]. Evidence also suggests that physicians' practices are dynamic, with the type and number of services they provide changing throughout their career [8]. This longitudinal variation creates an even more challenging environment for HHRP. Although previous studies have provided evidence for the varying scope of practice offered by different family physicians in Canada, these mostly survey-based studies have suffered from low response rates, a biased population sample and recall bias when estimating the number and type of services being provided [5]. For example, the response rate from a Canadian physician workforce survey in 2001 was approximately 50% [5], subsequent surveys in 2004 and 2007 had response rates of 30–35% [9] and finally a recent Canadian Medical Association workforce survey in 2017 and 2019 had a response rate of 12% [10, 11].

Variation in physicians' scope of practice is a particularly prominent challenge for HHRP in the province of Ontario, which contains both high-density metropolitan areas and a large number of rural communities. The discrepancy in provincial healthcare access is highlighted by the fact that 18% of Ontario's population lives in rural areas but only 9.3% of family physicians serve in these areas [4, 12]. In addition, a much smaller proportion of specialists practice in rural communities as compared to urban centres. Mian et al. (2017) described the unique challenges in recruiting physicians to rural communities in northern Ontario [13]. For rural populations, the lack of access to primary care can lead to higher levels of chronic disease, lower life expectancy, and overall poorer health status [14–16]. For example, in rural areas the mortality due to respiratory and circulatory disease for both sexes was 10% higher, women had a 16% higher chance of mortality from diabetes and the life expectancy of males in rural areas was 2.8 years less than urban males [15].

Given the importance of family physicians to population health, we assessed the breadth of services provided by family physicians in Ontario. We used physician demographics, practice location, and billing data to compare physician practices across different regions based on geographic location (southern vs. northern Ontario) and rurality (rural vs. urban). To avoid the challenges of survey-based responses, we accessed administrative data through the Data and Analytics Service at ICES and identified 18 general services provided by family physicians. Overall, we examine a larger number of services than previous studies and, unlike previous work, we also consider the physician's stage of practice.

## Methods

### Setting

The data for this study covers the province of Ontario, Canada. Ontario is approximately 909,000 km$^2$ in size with a population of 14.2 million people [17]. Although northern Ontario constitutes 87% of the total provincial land area, it only contains 6% of the provincial

population. For health care management purposes, the province was geographically divided into 14 Local Health Integration Networks (LHIN). Although the regional delivery of health care services is now managed by Ontario Health Teams, we retain the LHIN terminology and geographic structure because it reflects the system in place at the time our data was collected.

## Data sources

We retrieved Ontario's 2017 administrative healthcare data from the Data and Analytics Service at ICES via an encrypted server. The data were anonymized, and individual physicians were not identifiable. The two primary data sources from which we retrieved data were the ICES Physician Database (IPDB) and the Ontario Health Insurance Plan (OHIP) billing database. The IPDB contains each physician's demographic information, such as sex and age, as well as their location of practice (i.e., LHIN, sub-LHIN, and rurality), years in practice, billing specialty, full-time equivalent status, and certifications. The OHIP billing database contains billing codes submitted to the provincial government by physicians to receive remuneration for the services provided to each patient. For example, when a physician assesses a patient within their clinic, they may bill the government using the A007 code while a physician in the emergency department may use the H133 code. Although specific billing codes can vary among physicians and patient interactions based on factors such as the setting in which care was provided, the billing code database nevertheless indicates the general health services provided to a patient by the physician.

## Defining the physician cohort

The complete IPDB covers the approximately 30,000 Ontario physicians who provide any type of medical service. To subset this database to only include family physicians, we first used the OHIP billing specialty code variable and selected physicians who were classified as "family practice and general practice". Although some of these physicians are listed as family medicine or general practitioner in the OHIP specialty codes, their primary billings were that of a specialist. This likely occurred for physicians that hold dual certification where family medicine or general practice was their first specialty, therefore these physicians were removed from the analysis cohort. To ensure that a comprehensive physician population was used in the analysis, we analyzed the billing codes for all physicians in the IPDB with missing data for their specialty or certification.

## Defining sentinel billing codes

Based on the existing literature, our analysis of physician billing codes, and an examination of the Ontario Schedule of Benefits for physicians [18], we classified the OHIP billing codes into 18 broad services that family physicians could be providing. These services were clinic, anaesthesia, emergency medicine, hospital medicine, home visits, mental health, long-term care, obstetrics, palliative care and medical assistance in dying, surgical assisting, chemotherapy administration, sports medicine, chronic pain, care of the elderly, addictions medicine, endoscopy, allergy, and sleep medicine. For each service, we created a list of billing codes that were unique to that service and could therefore be used to define whether a physician provided that service. For example, if a physician billed code H103, H133, or H153 in addition to several other codes for an emergency department assessment, that patient visit was categorized as an 'emergency medicine' service. The full list of these "sentinel" billing codes can be found in Table 1 below:

## Determining the services provided by physicians

To determine the range of services provided by each physician, we grouped all sentinel billing codes from each physician into their associated services and calculated each physician's total number of patient interactions for each service. We then converted these sums to a binary variable indicating whether a physician provided a service. To represent only the services that each physician provided consistently, rather than for occasional or one-off patient visits, we set a minimum threshold of patient interactions for each service. For example, a physician who billed fewer than 50 patient encounters in the emergency department would not be considered as providing emergency medicine services given the low number of encounters. Thresholds for each service were defined using the billing data, following an approach similar to previous studies [8, 19, 20]. These thresholds are provided in Table 1. The final dataset had one row per physician, with columns containing physician demographic data, certification, geographical location, and a series of binary variables for each of the 18 service categories.

## Analysis

For our analysis, we grouped the physician and services data into four regions defined by the matrix of geographic area (north vs. south) and rurality (rural vs. urban). Northern Ontario was defined based on the former boundaries of the Northwest and Northeast LHINs, as in previous studies [4]. The rural/urban classification for each physician was determined based on the rurality index of Ontario (RIO) [21]. This score provides a measure of rurality based on population size as well as the required travel time to both local and larger referral centers. Physicians were assigned the RIO score by ICES based on their primary practice, and any

**Table 1. Sentinel billing codes from Ontario and thresholds for each service that are used to determine if a physician provided a service.**

| | Service | Sentinel Billing Codes | Threshold (per year) |
|---|---|---|---|
| 1 | Clinic (CL) | A001, A003, A004, A007, A008, A005, A006, Q200, A911, A912, G700, G845, G847, G373, G590 | 50 visits |
| 2 | Anaesthesia (AN) | S205C, S206C, S207C, S287C, S323C, S332C, S329C, S339C, S340C, S321C, S165C, S166C, S167C, S171C, S175C, S176C, S177C, S188C, E022C, E023C, P018C, P014C, P016C | 10 procedures |
| 3 | Emergency Medicine (EM) | H101, H102, H103, H151, H152, H153, H121, H122, H123, H131, H132, H133, A888, A100 | 50 visits |
| 4 | Hospital Medicine (HM) | C002, C003, C004, C007, C009, C008, C122, C123, C124, C911, C912, C933, A933 | 25 visits |
| 5 | Home Visits (HV) | A900, A901, A902 | 10 visits |
| 6 | Mental Health Services (MH) | K004, K007, K010, K012, K013, K019, K020, K024, K025, K040, K041, K005, K006 | 25 visits |
| 7 | Long-term Care (LT) | W001, W002, W003, W004, W008, W010, W105, W008 | 25 visits |
| 8 | Obstetrics (OB) | P006, P009 (vaginal delivery) | 2 deliveries |
| 9 | Palliative Care and MAID (PC) | C882, A945, K023, B997, B998, B966, G512, C945, E083, E084 | 10 visits |
| 10 | Surgical Assisting (SA) | Any fee code with the suffix B | 10 procedures |
| 11 | Chemotherapy Administration (CA) | G381, G345, G359 | 10 visits |
| 12 | Sports Medicine (SM) | A917 | 25 visits |
| 13 | Chronic Pain (CP) | A937, K707, K037 | 25 visits |
| 14 | Care of the Elderly (CE) | A967 | 25 visits |
| 15 | Addictions Medicine (AM) | K682, K683, K684, A957, A680, C680, K680 | 25 visits |
| 16 | Endoscopy (EN) | G379, Z399, Z400, E702, E797, E798, G379, Z491-Z499, Z555, E740, E741, E747, E705, Z571, E720, Z580, | 10 visits |
| 17 | Allergy (AL) | A927 | 25 visits |
| 18 | Sleep Medicine (SL) | A947 | 25 visits |

physician with a RIO score ≥40 was classified as working in a rural community [22]. This cut-off score is commonly used in ICES studies [23].

Using our data on physician demographics and provided services, we examined the differences in physician practice between rural and urban areas in northern and southern Ontario. We visualized the patient services landscape by producing an *UpSet* plot for each of the four study regions (north-urban, north-rural, south-urban, south-rural). Each plot showed the combination of services provided in a particular region, the proportion of physicians providing that combination of services, and the proportion of physicians with different ranges of years of practice experience [24, 25]. For clarity, each plot only shows the six most provided services in our data. All plots were created using R version 3.3.0 [26].

### Ethics approval

The Research Ethics Board at Lakehead University approved this study (#1466634).

## Results

Our data encompasses 14,443 family physicians. Most of the family physicians in our data practiced in urban settings in southern Ontario (Table 2). Between 40 and 47% of physicians were female, depending on the region. Physician years in practice varied across the four regions, with the north-rural region having the highest proportion of physicians in their first ten years of practice. Conversely, the other three regions had higher proportions of physicians with more than 30 years of experience. The six most common healthcare services provided by physicians across all four regions were clinic practice, mental health, emergency medicine, palliative care, hospital medicine and home visits. Although these services were most common within the four regions, the proportion of physicians practicing these services varied by region.

We first examined the number of services provided by family physicians among the four study regions (Fig 1). In general, physicians in the south-urban region provided a smaller number of services than the other three regions. Furthermore, physicians in their first decade of practice typically provided a greater number of services than more experienced physicians, a trend that was consistent across all four regions (Fig 1).

In the **north-rural** region (Fig 2), just under 25% of physicians provided the full combination of clinic, hospital medicine, emergency medicine, and palliative care services. Another 20% practiced mental health in addition to these four services (Fig 2). Notably, less than 5% of physicians practiced all six services shown in Fig 2 and they are distributed relatively equal amongst the range of years in practice.

As in the north-rural region, approximately 25% of physicians in the **north-urban** region practiced in four to five of the most common services (Fig 3). In addition, a moderate proportion of physicians provided one to three services with the clinic, hospital medicine, and emergency medicine services being the most common. The physicians who provided these narrower combinations of services generally had more years in practice (> 30) relative to those who provided a greater range of services.

In the **south-rural** region, the four most common combinations of provided services included four to six services and represented approximately 43% of physicians (Fig 4). As in the north-urban region, physicians in the next four combinations of services only provided between one and three services.

The **south-urban** region showed a different pattern of practice relative to the other three regions (Fig 5). Here, the most common combinations of provided services were limited to a narrower range of one to three services, with a smaller proportion of the physicians providing a combination of four to six services.

**Table 2. Description of physician demographics, years of experience, number, and type of service by region in Ontario.**

| | North-Rural | North-Urban | South-Rural | South-Urban | Total |
|---|---|---|---|---|---|
| Total Physicians | 293 | 545 | 759 | 11387 | 12984 |
| Total Females | 118 (40%) | 235 (43%) | 328 (43%) | 5300 (47%) | 5981 (46%) |
| **Years of Experience** | | | | | |
| <10 | 144 (49%) | 233 (43%) | 333 (44%) | 4243 (37%) | 4953 (38%) |
| 10–19 | 55 (19%) | 91 (17%) | 126 (17%) | 2172 (19%) | 2444 (19%) |
| 20–29 | 70 (24%) | 132 (24%) | 190 (25%) | 3337 (29%) | 3729 (29%) |
| >29 | 24 (8%) | 89 (16%) | 110 (14%) | 1635 (14%) | 1858 (14%) |
| **Services Provided** | | | | | |
| Clinic | 259 (88%) | 439 (81%) | 666 (88%) | 9838 (86%) | 11202 (86%) |
| Mental Health | 113 (39%) | 299 (55%) | 511 (67%) | 7737 (68%) | 8660 (67%) |
| Emergency Medicine | 197 (67%) | 305 (56%) | 483 (64%) | 6511 (57%) | 7496 (58%) |
| Palliative Care | 196 (67%) | 282 (52%) | 502 (66%) | 2897 (25%) | 3877 (30%) |
| Hospital Medicine | 224 (76%) | 280 (51%) | 452 (60%) | 1936 (17%) | 2892 (22%) |
| Home Visits | 27 (9%) | 72 (13%) | 206 (27%) | 1835 (16%) | 2140 (16%) |
| Long-term Care | 122 (42%) | 65 (12%) | 285 (38%) | 1258 (11%) | 1730 (13%) |
| Chronic Pain | ≤6 (0%)* | 31 (6%) | 47 (6%) | 883 (8%) | 962–967 (7%) |
| Surgical Assisting | 26 (9%) | 51 (9%) | 106 (14%) | 819 (7%) | 1002 (8%) |
| Anaesthesia | 17 (6%) | 38 (7%) | 58 (8%) | 532 (5%) | 645 (5%) |
| Obstetric Deliveries | 26 (9%) | 16 (3%) | 39 (5%) | 244 (2%) | 325 (3%) |
| Addiction Medicine | 0 | 16 (3%) | < = 6 (0%) | 222 (2%) | 239–244 (2%) |
| Endoscopy | 26 (9%) | < = 6 (0%) | 24 (3%) | 98 (1%) | 149–154 (1%) |
| Sports Medicine | 0 | 0 | 0 | 74 (1%) | 74 (1%) |
| Chemotherapy Administration | ≤6 (0%) | ≤6 (1%) | ≤6 (1%) | 15 (0%) | 18–33% (0%) |
| Care of the Elderly | 0 | 0 | 0 | 0 | 0 |
| Allergy Medicine | 0 | 0 | 0 | 0 | 0 |
| Sleep Medicine | 0 | 0 | 0 | 0 | 0 |
| **Number of Services provided** | | | | | |
| 1 | 38 (13%) | 95 (17%) | 68 (9%) | 1544 (14%) | 1745 (13%) |
| 2 | 28 (10%) | 94 (17%) | 83 (11%) | 2943 (26%) | 3148 (24%) |
| 3 | 29 (10%) | 88 (16%) | 117 (15%) | 3333 (29%) | 3567 (27%) |
| 4 | 59 (20%) | 98 (18%) | 106 (14%) | 1688 (15%) | 1951 (15%) |
| 5 | 62 (21%) | 97 (18%) | 122 (16%) | 1033 (9%) | 1314 (10%) |
| 6 | 41 (14%) | 45 (8%) | 125 (16%) | 505 (4%) | 716 (6%) |
| 7 | 24 (8%) | 20 (4%) | 93 (12%) | 227 (2%) | 364 (3%) |
| > = 8 | 12 (4%) | 8 (1%) | 45 (6%) | 114 (1%) | 179 (1%) |

*To ensure anonymity for physicians, ICES requires all small sizes (i.e., ≤6) to be suppressed. Row totals will be expressed as a range.

## Discussion

### Summary of findings

Our results represent one of the most comprehensive descriptions of the services provided by Ontario family physicians in relation to geography, rurality, and number of years in practice. In addition, our data demonstrates the broad scope of practice for family physicians in Ontario. These observations are similar to previous studies that demonstrated that rural generalist physicians practice to a fuller extent of their scope of practice [4, 5]. However, our work extends these previous observations by identifying the frequency of different combinations of

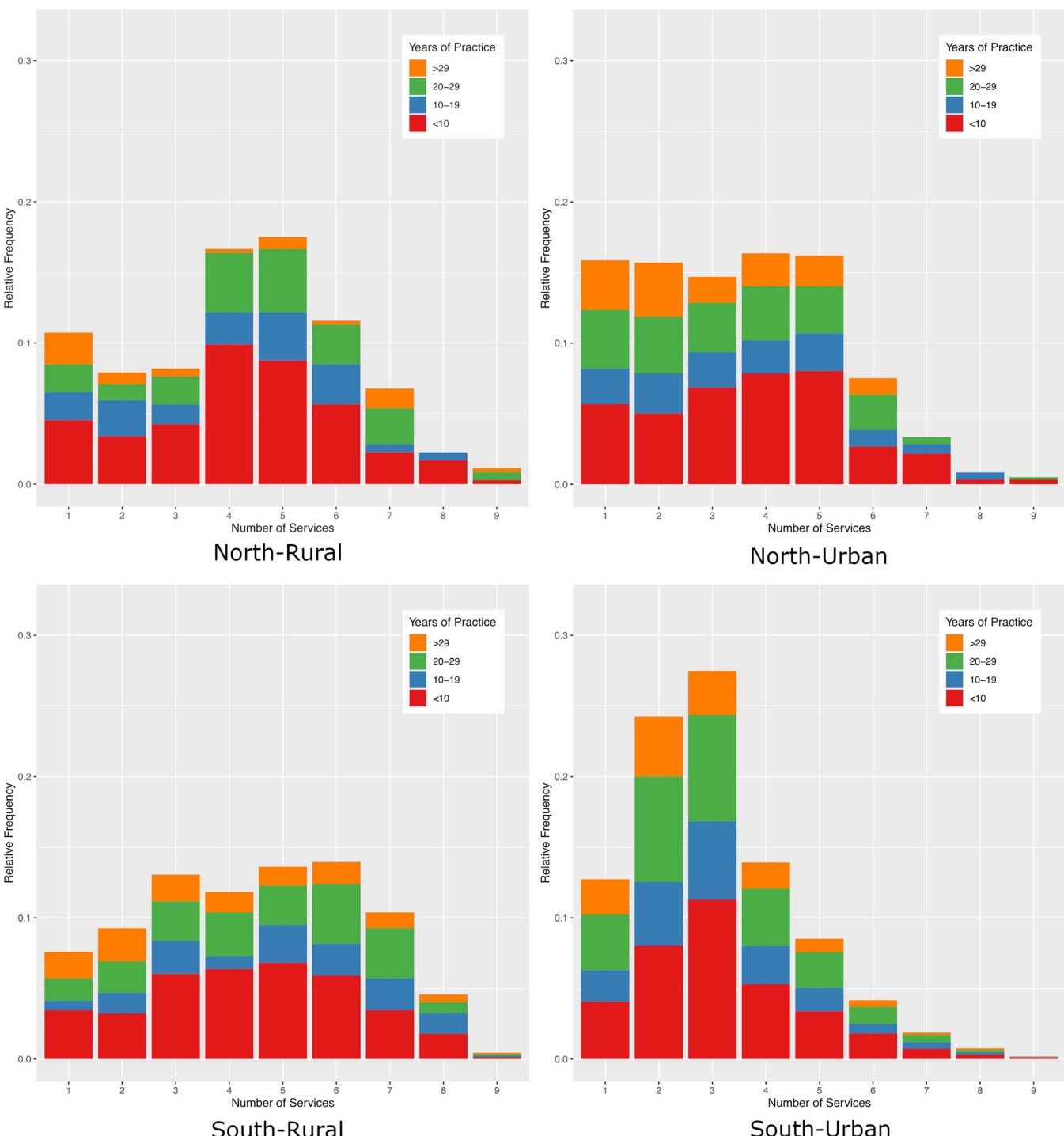

**Fig 1. A frequency plot showing the proportion of physicians by number of services classified by their years of experience.** The data is presented for all four regions of interest.

the most common services provided by family physicians. Overall, the six most common services provided by family physicians in Ontario were clinic services, palliative care, hospital medicine, emergency medicine, mental health, and home visits, though we found that physicians provided these services in different combinations and frequencies among our four study

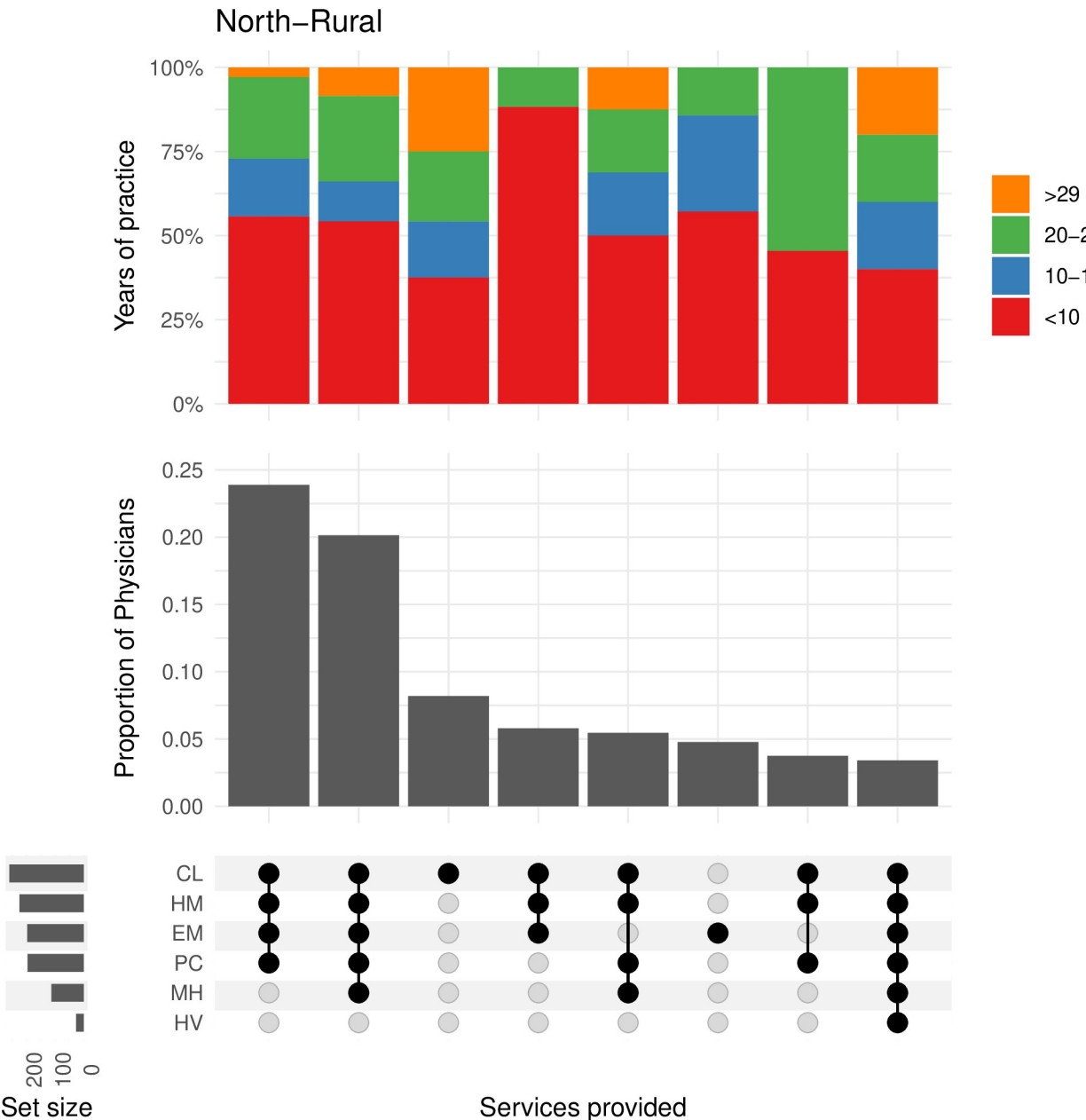

**Fig 2. This plot shows the frequency of physicians and the combination of services they are providing in the north-rural region.** The bar plot above shows the proportion of physicians by their years of experience within each of the combinations in services. CL = Clinic, HM = Hospital Medicine, EM = Emergency Medicine, PC = Palliative Care, MH = Mental Health, HV = Home Visits.

regions. We also identified the number of years in practice for the physicians practicing each of the provided service combinations.

## Previous research

Previous studies have taken different approaches to identifying the services provided by family physicians: where some focused on the range of different services [4, 5], others focused on the comprehensiveness of the provided care [19]. Here, we used the OHIP Schedule of Benefits to

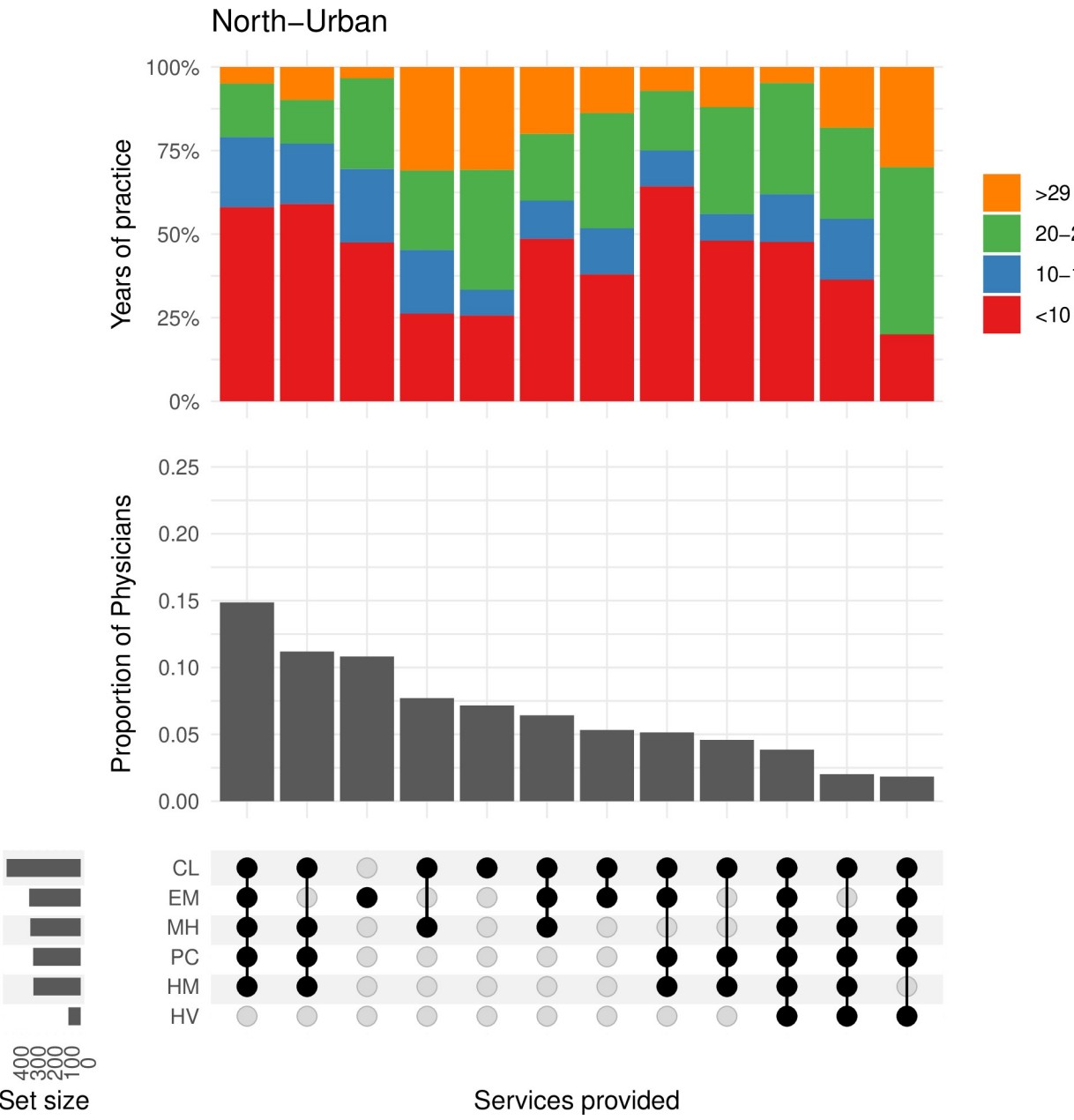

**Fig 3. This plot shows the frequency of physicians and the combination of services they are providing in the north-urban region.** The bar plot above shows the proportion of physicians by their years of experience. The bar plot above shows the proportion of physicians by their years of experience within each of the combinations in services. CL = Clinic Medicine, HM = Hospital Medicine, EM = Emergency Medicine, PC = Palliative Care, MH = Mental Health, HV = Home Visits.

identify a total of 18 different categories of physician services [18]. Unlike the Ontario-based studies by Schultz and Glazier [19] and Chan [27], which only considered comprehensive family physicians providing primary care, we examined the physician workforce from an HHRP perspective by including all family physicians (i.e., even those with a focused practice). The variability that we observed with respect to the services provided across our four study regions agrees with previous results from Wong and Stewart (2010) [5].

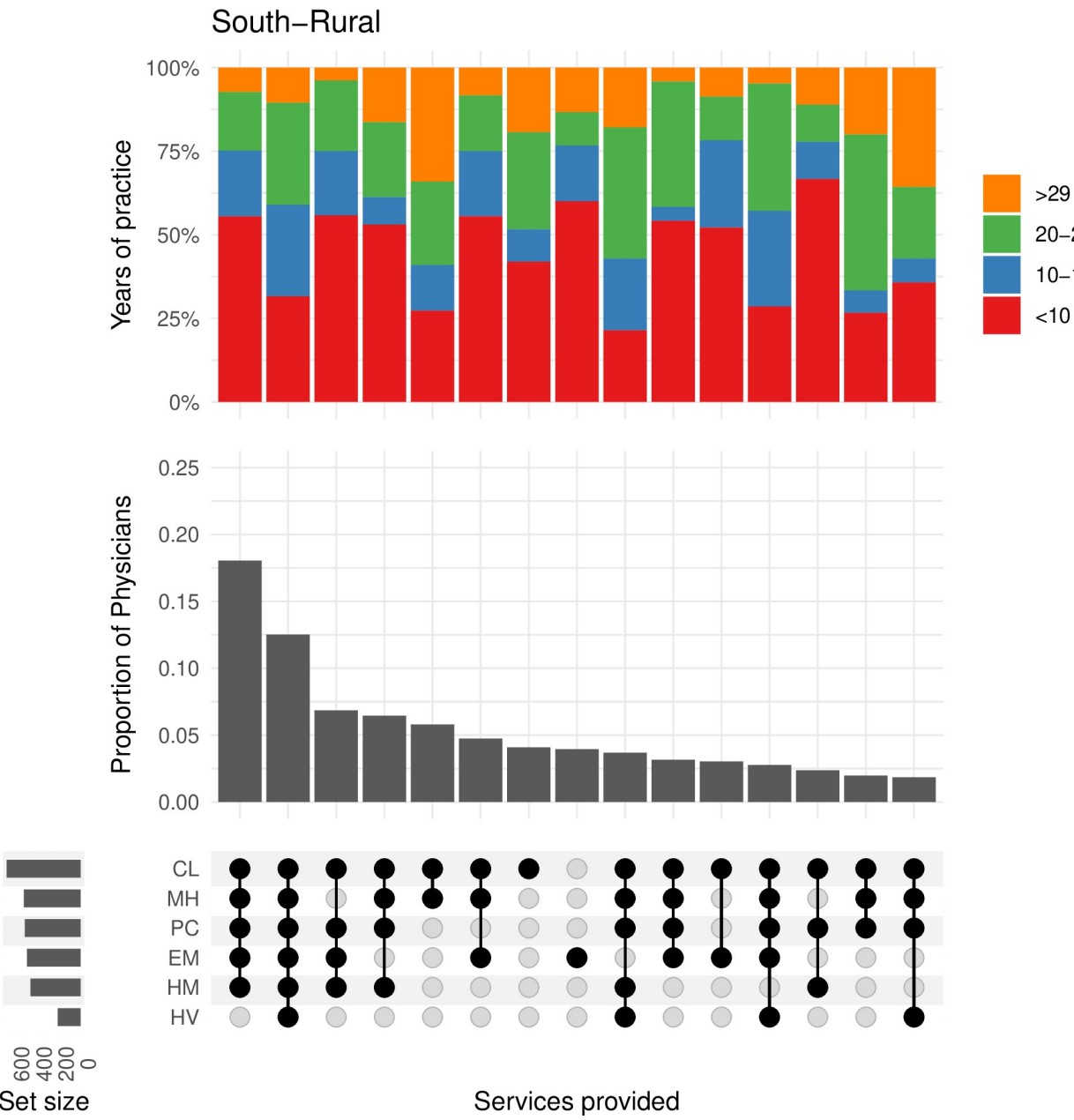

**Fig 4. This plot shows the frequency of physicians and the combination of services they are providing in the south-rural region.** The bar plot above shows the proportion of physicians by their years of experience within each of the combinations in services. CL = Clinic Medicine, HM = Hospital Medicine, EM = Emergency Medicine, PC = Palliative Care, MH = Mental Health, HV = Home Visits.

### The value of physician-specific scope of practice data

By evaluating the more granular details of the specific number and types of services provided by family physicians, our results will support future reality-based forecasting and more robust HHRP in Ontario and elsewhere. For example, Ontario performed HHRP in the mid-2000s and concluded that the province would have a sufficient number of family physicians by 2018 [28]. However, this evaluation made incorrect assumptions about family physician services, including the assumption that family physician services are all clinic-based. As a result,

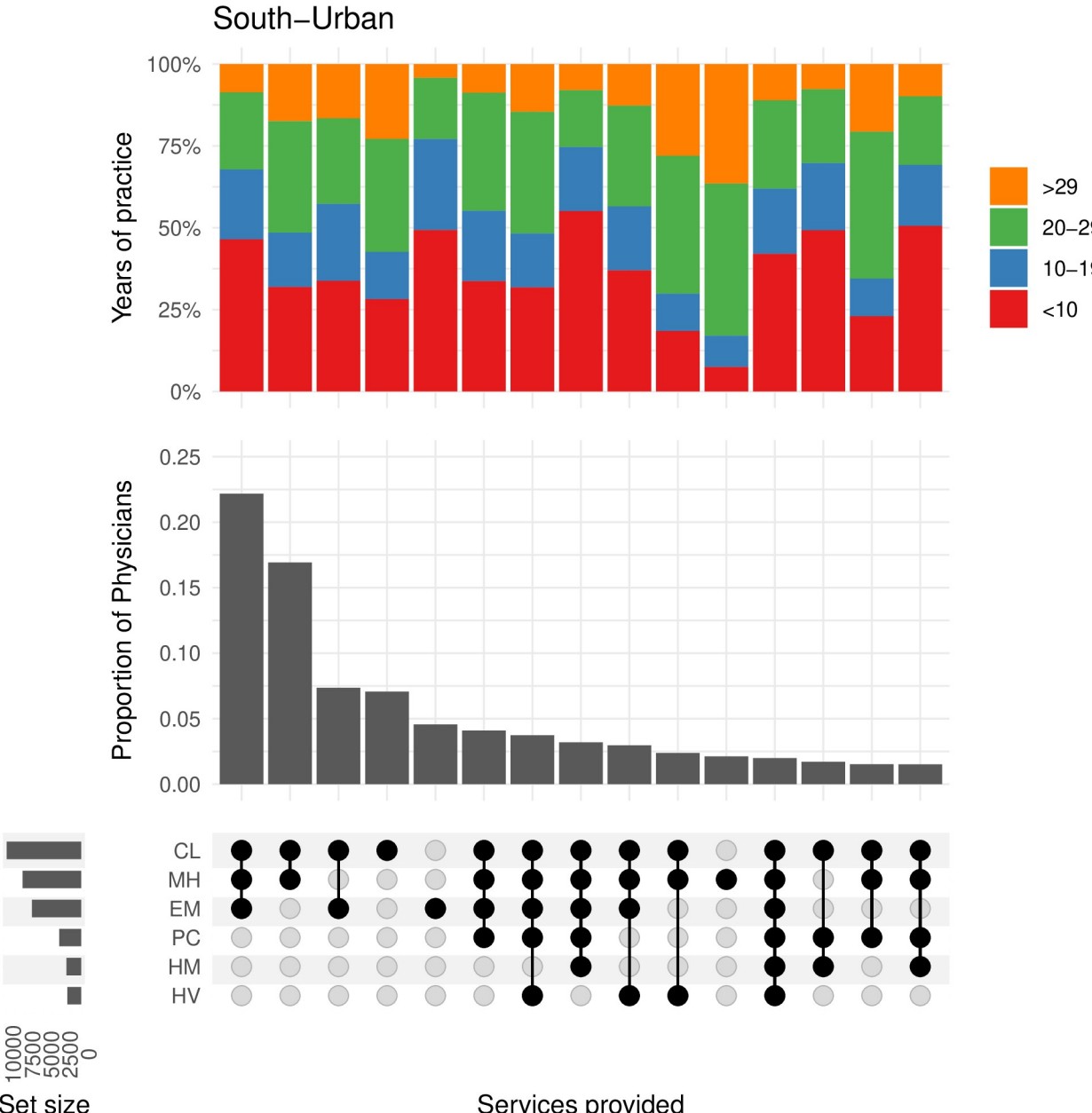

**Fig 5. This plot shows the frequency of physicians and the combination of services they are providing in the south-urban region.** The bar plot above shows the proportion of physicians by their years of experience within each of the combinations in services. CL = Clinic Medicine, HM = Hospital Medicine, EM = Emergency Medicine, PC = Palliative Care, MH = Mental Health, HV = Home Visits.

Ontario has been facing a shortage of physicians able to provide services such as emergency medicine and hospital medicine. More recent studies have shown that many family physicians have diverse practices that include more than just clinic medicine [4, 5, 8, 19, 20].

Our results also provide valuable information that could be used to shape medical education. Although all family medicine residents in Canada are required to meet the same set of competencies, our results provide valuable information on the knowledge and skills that family physicians will need based on the region of Ontario in which they practice. Residents can use this information to pursue learning opportunities that will help them meet the needs of the

population in which they intend to serve. Furthermore, medical schools can use this information to customize their curriculum: for example, one medical school in Ontario has already developed a rural generalist pathway to deal with the severe shortage of physicians in many rural communities [29].

Finally, our data will help policymakers plan for ensuring future decades of clinical services across the province. Our results also underscore the diversity of physician practices and the extent to which practices can vary between northern and southern Ontario and between urban and rural communities. Like Simkin et al. (2019), our results also demonstrate how physicians' practices can change as they approach retirement [20]. In addition, the results show that a higher proportion of early career physicians practice in rural communities. Although the administrative data provides little insight into why this is occurring, Jolicoeur et al. (2022) found that physician retention in rural communities can be influenced by spousal employment opportunities, difficulty integrating into the community, physician burnout, and lack of resources [30]. Quantifying and understanding these differences is crucial for clinical service planning, especially for northern and rural communities where family physicians practice to greater extent of their scope of practice. The data from this study outlines the services already being delivered within different communities. Indeed, understanding the healthcare needs of a community requires an understanding of the number of physicians available, the type of care they can provide, and the number of patients they can service. Because many communities are already under-serviced, our findings should not be used to estimate demand, but they can nevertheless indicate whether a service should be offered within a community. Addressing these needs, however, will require developing new strategies to increase family physician recruitment and retention in rural Canada, where much of the historical emphasis has been on education, training, and financial incentives [6].

## Family medicine practice internationally

Western countries such as Canada and the United States have the most comprehensive scope of family medicine training, covering primary, emergency, hospital, long-term, maternal, and intrapartum care [31]. Australia's rural practice stream matches this breadth, but its general practice stream is more limited, excluding emergency and intrapartum care. New Zealand, the United Kingdom, and Ireland offer narrower clinical training, with less focus on hospital and emergency care, though they still emphasize primary care and long-term care [31]. In terms of physician training time, Scandinavian countries have longer training programs, while Canada and Lebanon, have shorter two-year residencies [32]. The analysis from this study provides a comprehensive approach for policy and decision makers to quantify family physician scope internationally.

## Limitations and future direction

Although our study provides a comprehensive analysis of the services provided by family physicians, our approach does not account for private services or those not billable by the OHIP (e.g., aesthetic medicine, coroner work, or occupational medicine). Furthermore, family physicians who provide services outside the typical fee-for-service model will often submit shadow billings to the government for a small proportion of their billings. If these physicians do not submit their shadow billings on a regular basis or in a particular setting that is represented by one or several of our sentinel billing codes, our assessment of the services they provide may be incomplete. Given these limitations, our analysis likely represents a lower bound estimate for the number and types of services provided across Ontario. We additionally note that the data used in this analysis comes from 2017, three years before the COVID pandemic. Although the

number of physicians practicing in Ontario has changed in the intervening years due to retirement and migration, we believe our analysis provides a thorough description of the family physician scope of practice in Ontario. This study also provides a methodology for future studies either in Ontario or other healthcare jurisdictions where health human resource planning is a challenge. Furthermore, longitudinal studies in Ontario could reveal changes in physician practices over time, which could inform HHRP for the future.

## Conclusions

Family physicians are essential for delivering primary care and thereby improving and promoting population health. Overall, this cross-sectional study provides important information about the range of services provided by Ontario family physicians in four distinct geographic areas and at different stages in their careers. These findings have the potential to contribute to HHRP, medical education, and clinical services planning in Ontario and elsewhere. This study provides policy- and decision-makers in Ontario and elsewhere with a basis for integrating knowledge of the specific clinical services delivered by family physicians into their planning, with the goal of ensuring a fit-for-purpose workforce able to meet community healthcare needs.

## Acknowledgments

This study contracted ICES Data & Analytic Services (DAS) and used de-identified data from the ICES Data Repository, which is managed by ICES with support from its funders and partners: Canada's Strategy for Patient-Oriented Research (SPOR), the Ontario SPOR Support Unit, the Canadian Institutes of Health Research and the Government of Ontario. The opinions, results and conclusions reported are those of the authors. No endorsement by ICES or any of its funders or partners is intended or should be inferred.

## Author Contributions

**Conceptualization:** David W. Savage, Roger Strasser.

**Data curation:** David W. Savage, Arunim Garg, Vijay Mago.

**Formal analysis:** David W. Savage, Arunim Garg, Salimur Choudhury, Vijay Mago.

**Funding acquisition:** David W. Savage, Roger Strasser.

**Investigation:** David W. Savage, Arunim Garg, Vijay Mago.

**Methodology:** David W. Savage, Arunim Garg, Vijay Mago.

**Project administration:** David W. Savage, Vijay Mago.

**Resources:** David W. Savage, Vijay Mago.

**Software:** Arunim Garg, Salimur Choudhury, Vijay Mago.

**Supervision:** David W. Savage, Salimur Choudhury, Vijay Mago.

**Validation:** David W. Savage, Arunim Garg, Salimur Choudhury, Vijay Mago.

**Visualization:** David W. Savage, Arunim Garg, Salimur Choudhury, Vijay Mago.

**Writing – original draft:** David W. Savage, Arunim Garg, Vijay Mago.

**Writing – review & editing:** David W. Savage, Arunim Garg, Salimur Choudhury, Roger Strasser, Robert Ohle, Vijay Mago.

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
