## [Decision Letter · Decision Letter 0]

3 Sep 2024

PONE-D-24-16194Characterizing the services provided by family physicians in Ontario, Canada: A retrospective study using administrative billing dataPLOS ONE

Dear Dr. Mago,

Thank you for submitting your manuscript to PLOS ONE. After careful consideration, we feel that it has merit but does not fully meet PLOS ONE’s publication criteria as it currently stands. Therefore, we invite you to submit a revised version of the manuscript that addresses the points raised during the review process.

Please take into consideration the remarks of the reviewers below.

We look forward to receiving your revised manuscript.

Kind regards,

Inge Roggen, M.D., Ph.D.

Academic Editor

PLOS ONE

“RS is a co-investigator with DS as the principal investigator on a grant from the Northern Ontario Academic Medical Association (NOAMA) Clinical Innovation Opportunities Fund (C-18-19). Additional funding to SC from the Natural Sciences and Engineering Research Council (NSERC) of Canada supported this work.”

“This study contracted ICES Data & Analytic Services (DAS) and used de-identified data from the ICES Data Repository, which is managed by ICES with support from its funders and partners: Canada’s Strategy for Patient-Oriented Research (SPOR), the Ontario SPOR Support Unit, the Canadian Institutes of Health Research and the Government of Ontario. The opinions, results and conclusions reported are those of the authors. No endorsement by ICES or any of its funders or partners is intended or should be inferred.”

“RS is a co-investigator with DS as the principal investigator on a grant from the Northern Ontario Academic Medical Association (NOAMA) Clinical Innovation Opportunities Fund (C-18-19). Additional funding to SC from the Natural Sciences and Engineering Research Council (NSERC) of Canada supported this work.”

Reviewers' comments:

Reviewer's Responses to Questions

**Comments to the Author**

1. Is the manuscript technically sound, and do the data support the conclusions?

Reviewer #1: Yes

Reviewer #2: Yes

2. Has the statistical analysis been performed appropriately and rigorously? 

Reviewer #1: Yes

Reviewer #2: Yes

3. Have the authors made all data underlying the findings in their manuscript fully available?

Reviewer #1: Yes

Reviewer #2: Yes

4. Is the manuscript presented in an intelligible fashion and written in standard English?

Reviewer #1: Yes

Reviewer #2: Yes

5. Review Comments to the Author

Reviewer #1: This is an excellent study: well planned, well analysed and well presented. It is of great importance for planning the education of family physicians and explaining to planners the exact situation now and in the future. My comments are only to improve the impact of their study.

Introduction

“Moreover, strong primary care systems have been shown to improve overall population health and reduce health system costs (2).”

[This is crucial to the importance and uptake of your research. Can you please provide perhaps from systematic reviews a summary that will convince readers]

“Although 69 previous studies have provided evidence for the varying scope of practice offered by different 70 family physicians in Canada, these mostly survey-based studies have suffered from low 71 response rates and recall bias when estimating the number and type of services being provided 72 (4”

[Can you provide examples of low response rates for important topics. This will greatly increase the importance of your study]

“In addition, a much smaller proportion of specialists practice in rural 79 communities as compared to urban centres. Mian et al. (2017) described the unique challenges 80 in recruiting physicians to rural communities in northern Ontario (9). For rural populations, the 81 lack of access to primary care can lead to higher levels of chronic disease, lower life expectancy, 82 and overall poorer health status (10–12).”

[This is the essence of the problem. Can you provide some supporting stats please?]

Data sources

“We retrieved Ontario’s 2017 administrative healthcare data”

[readers will wonder why 2017 and not 2023. Please explain]

Typos

107 data was anonymized, [please change to were. Latin datum = singular, data = plural]

139 in additional [change to in addition to]

193 provided a fewer number of [change to smaller]

204 Figure 2, however, the among different year in 205 practice in relatively equal [please rewrite]

I have published a study of readmission rates and mortality of seniors admitted to the Calgary hospitals 2013-2023. I sent it to the Premier, Health Minister and all the physician organisations. I was advised that the bureaucrats are the key persons who might take action on it.

Reviewer #2: Well perfomed analysis of your local situation of services by family physicians. Unfortunately, it is very difficult to understand details for persons who are not familiar with your billing system. What do all these codes mean? Why is there such a low number of items for chonic care? What is about cooperations within one practice setting? Are they complementary? Why is the number of young practicing doctors higher in rural regions? Are many of them moving to urban areas after some years? If you wsh to publish your data in an international data you should give much more informations in the appendix and you should draw international comparisons

6. PLOS authors have the option to publish the peer review history of their article (what does this mean?). If published, this will include your full peer review and any attached files.

Reviewer #1: **Yes: **Roger E. Thomas

Reviewer #2: **Yes: **Prof. Dr. Erik Baum, Germany

---

## [Author Response · Author response to Decision Letter 0]

13 Nov 2024

A response letter has been added in the revised submission

---

## [Decision Letter · Decision Letter 1]

13 Dec 2024

Characterizing the services provided by family physicians in Ontario, Canada: A retrospective study using administrative billing data

PONE-D-24-16194R1

Dear Dr. Mago,

We’re pleased to inform you that your manuscript has been judged scientifically suitable for publication and will be formally accepted for publication once it meets all outstanding technical requirements.

Kind regards,

Inge Roggen, M.D., Ph.D.

Academic Editor

PLOS ONE

Additional Editor Comments (optional):

Reviewers' comments:

Reviewer's Responses to Questions

**Comments to the Author**

1. If the authors have adequately addressed your comments raised in a previous round of review and you feel that this manuscript is now acceptable for publication, you may indicate that here to bypass the “Comments to the Author” section, enter your conflict of interest statement in the “Confidential to Editor” section, and submit your "Accept" recommendation.

Reviewer #1: All comments have been addressed

Reviewer #2: All comments have been addressed

2. Is the manuscript technically sound, and do the data support the conclusions?

Reviewer #1: Yes

Reviewer #2: Yes

3. Has the statistical analysis been performed appropriately and rigorously? 

Reviewer #1: Yes

Reviewer #2: Yes

4. Have the authors made all data underlying the findings in their manuscript fully available?

Reviewer #1: Yes

Reviewer #2: Yes

5. Is the manuscript presented in an intelligible fashion and written in standard English?

Reviewer #1: Yes

Reviewer #2: Yes

6. Review Comments to the Author

Reviewer #1: The authors have carefully replied to all of the reviewers' comments and questions and fully explained why they took particular strategies in the data analysis. They discussed the limitations of the data in their databases. Important and thoughtfully planned, researched and analysed and presented study on an important topic of concern to family physicians, primary care planners, primary care researchers and hopefully ministries and funding organisations. The databases is appropriate to the analysis of these concerns. The authors provided careful and thorough responses to the reviewers' comments and satisfactorily explained their strategies and decisions about the data. The authors carefully explained the data privacy requirements and why certain data cannot be disclosed in this publication. They explained why nurse practitioners' salaries are not available for publication in this article. They also explained that different billing codes would be used by a family physician practising in their office and a family physician practising in an emergency ward. They explained why the 2017 and not later dataset was used. Good discussion of differences by age between rural and urban physicians in their practices. Clearly written and analysed. Good graphics clearly presented their data and which are very easy to comprehend. The manuscript is technically sound and the analysis, discussion and graphical presentations support their conclusions.

Reviewer #2: requests for corrections were adressed adequately. Now international aspects and reader for codes are added. I have no further comments

7. PLOS authors have the option to publish the peer review history of their article (what does this mean?). If published, this will include your full peer review and any attached files.

Reviewer #1: **Yes: **Roger E.Thomas

Reviewer #2: No

---

## [Editor Report · Acceptance letter]

27 Dec 2024

PONE-D-24-16194R1 

PLOS ONE

Dear Dr. Mago, 

I'm pleased to inform you that your manuscript has been deemed suitable for publication in PLOS ONE. Congratulations! Your manuscript is now being handed over to our production team.

Kind regards, 

on behalf of

Prof. Inge Roggen 

Academic Editor

PLOS ONE